# Ghosting: Abandonment in the Digital Era

**Lateefa Rashed Daraj [1], Mariam Rashid Buhejji [1], Gretta Perlmutter [2], Haitham Jahrami [1,3,*] and Mary V. Seeman [4]**

[1]   Department of Psychiatry, College of Medicine and Medical Sciences, Arabian Gulf University, Manama P.O. Box 26671, Bahrain; latifarashiddarraj12@gmail.com (L.R.D.); maryamrmb@agu.edu.bh (M.R.B.)
[2]   Coping with Ghosting LLC, San Diego, CA 92037, USA; copingwithghosting@gmail.com
[3]   Government Hospitals, Manama P.O. Box 5128, Bahrain
[4]   Department of Psychiatry, University of Toronto, Toronto, ON M5S 1A8, Canada; mary.seeman@utoronto.ca
[*]   Correspondence: haitham.jahrami@outlook.com

**Definition:** This entry synthesizes the multidisciplinary literature on ghosting published through late 2023 across psychological and social science journals. Search terms include "ghosting" and "online dating". Both quantitative and qualitative studies are included. The rise in ghosting can be attributed to advancements in technology and the increased popularity of dating apps. It is defined as an abrupt one-sided ending, without explanation, of an established friendship/romantic or other communication connection. The prevalence of ghosting has increased, as reported by both ghosters (i.e., persons who stopped responding) and ghostees (i.e., persons who were "dumped"). Individuals characterized by dark triad traits (i.e., psychopathy, Machiavellianism, and narcissism) are more likely than others to be ghosters. These individuals have a history of using ghosting as their preferred method of ending relationships without concern for its negative impact on ghostees or, indeed, on themselves. The psychological effects of ghosting can influence mental health, although most individuals ultimately find ways of coping.

**Keywords:** breadcrumbing; ghostee; ghoster; ghosting; initiator; online dating; terminator

## 1. Introduction

Although never easy, there are various ways, from protective to cruel, of ending a relationship. These can vary from gradual to abrupt and from face-to-face discussion and in-depth explanation to silence and avoidance [1,2]. An extreme form of avoidance is called ghosting [2]. Ghosting is characterized by the sudden and unexpected termination of all forms of communication without a clear explanation or advanced discussion. This can happen either suddenly or gradually [1,3,4]. It usually involves the act of "unfriending" or "unmatching" a person on social media [1,3], but it can be done through nonresponse to phone calls, emails, or text messages [4]. LeFebvre and colleagues, as well as Koessler and colleagues, argued that ghosting is limited to romantic relationships [5,6]. However, recent studies based on insights and interviews have demonstrated that this phenomenon also occurs in the context of friendships [7–9]. A unique study performed by Forrai et al. further supported this contention, showing that ghosting a romantic partner is different than ghosting a friend. It was found that communication overload (i.e., the sense of being overwhelmed by the frequency of messaging) was a predictor of ghosting romantic partners, while self-esteem issues predicted ghosting friends. Additionally, the evidence suggests that ghosting within romantic relationships does not suffer long-term effects, whereas friend ghosting leads to depressive symptoms and has negative consequences for both the ghoster and the ghostee [8].

The emergence of ghosting must be understood within the shifting of historical dynamics between direct person-to-person relating and the new mode of relating via communication media [5]. In the past, face-to-face meetings built and strengthened social ties, making each party accountable for behavior during contact [10]. The advent of the telegraph

and telephone subsequently permitted communication during physical separations, thus maintaining relationships over a distance but reducing closeness [11]. Radio and television centralized one-way intimate broadcasts into homes, creating fantasy relationships that, at times, felt real [4]. Internet forums, emailing, and text messaging allowed fast, low-cost, asynchronous connections between people over a geographical distance, spurring a form of disembodied intimacy among relative strangers [5]. Social platforms have expanded this form of pseudo-intimacy [9,12]. Each advancement, perhaps paradoxically, made it easier to avoid the difficult conversations that inevitably arise over time between persons building a potentially intimate relationship [10]. The present-day practice of ghosting thus reflects what happens when attempting to relate through impersonal means rather than face-to-face. It takes advantage of the impersonal to avoid painful conversations and social accountability.

The widespread use of social media and cell phones has made ghosting the path of least resistance [13], the motive being wanting to end a relationship without the discomfort of a face-to-face meeting [4]. The precursors are finding communications with the ghostee to be annoying, boring, overly intrusive, and time-consuming or to be obstacles to the formation of other preferred relationships [13]. A study conducted by Timmermans et al. revealed three themes when reviewing ghostees' perspectives on reasons for being ghosted. Approximately 59% blamed the ghoster, attributing his/her actions to personality flaws such as cowardice or unresolved commitment issues or other relationship breakdowns, such as loss of romantic interest or involvement in a new relationship. Another 37% of the participants placed the blame on themselves and were convinced that they had said or done something wrong or were simply unworthy of the relationship. A third (overlapping) group of 37% blamed dating apps that facilitated ghosting [12]. Dating apps are the latest communication technology to affect relationships. Features such as swiping and messaging foster quick superficial judgments of the persons with whom one "talks". This makes connections relatively remote and fosters the emergence of ghosting, seemingly painless breakups [12].

The same study also examined the reasons given by ghosters and identified five major themes. Three were the same as those of ghostees. In addition, some felt that there was no obligation to discuss breaks beforehand, while others believed (or said they believed) that no explanation was less painful for the ghostee than the trauma of explanatory discussions [14].

With advances in technology and the evolution of social media, the prevalence of ghosting has increased, giving rise to a variety of online communication apps [4]. It has been reported that 20–40% of the general population has experienced ghosting either as a ghostee, a ghoster, or both [14,15]. The prevalence of having been ghosted by a romantic partner has been estimated to range between 13% and 23% for adults reporting the experience of having been ghosted by a romantic partner [15]. A survey of 1000 U.S. adults by YouGov and Huffington Post showed that 13% endorsed having been ghosted and 11% endorsed having been a ghoster [16]. Freedman et al., studying 554 U.S. adults, reported that 25.3% had been ghosted, whereas 21.3% had ghosted a former partner [2,17]. A second sample of 747 participants revealed that 23% had been ghosted and 18.9% had ghosted [17]. Similarly, LeFebvre et al. sampled 99 U.S. university students and reported that 29.3% of them ghosted, while 25.3% characterized themselves as ghostees [5]. In another recent study of 333 U.S. adults conducted by Koessler et al., 64.5% had ghosted, whereas 72% had been ghosted [18]. In Spain, 19.3% had experienced ghosting at least once in the year prior to a ghosting study [15]. Two studies concluded that ghosting was used at least one third of the time as the preferred rejection technique for dating apps [19,20]. Figure 1 provides the definition, prevalence, and impacts of ghosting.

# GHOSTING - Abandonment in the Digital Area

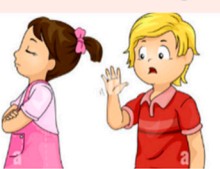

## How common is ghosting?

- **20 – 24%** have been a ghostee, ghoster, or both.

- **13 – 23%** of adults have been ghosted by a romantic partner.

- **1/3rd** of the time, ghosting is the preferred rejection technique on dating apps.

## What is the impact of ghosting?

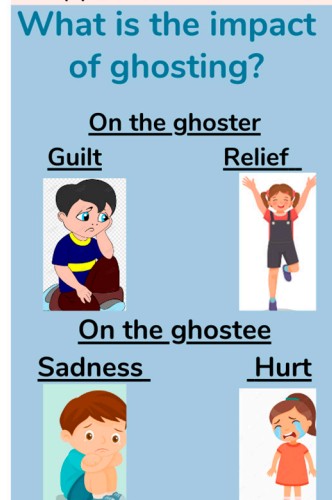

## What is ghosting?

It is the sudden and unexpected termination of all forms of communication with someone without a clear explanation or advance warning.

## Why is ghosting done?

1- To avoid the discomfort of face-to-face meeting.

2- Out of boredom and annoyance.

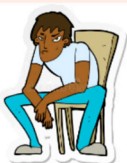

**Figure 1.** Summary of the important points pertaining to all forms of ghosting.

Ghosting is a topic of psychological interest because it occurs frequently, can cause mental distress, and has been little researched. Recently, a measure of ghosting (GHOST) based on the Shannon–Weaver communication model was developed to explore the experiences of ghostees [4]. The act of ghosting often exposes hidden dimensions of personality and is, for that reason, of interest to mental health researchers and providers. This approach may become a fruitful area of inquiry in the course of a mental health assessment [12,13].

Our central research question in this entry examines ghosting motivations and their effects on the persons involved. We propose that technologically induced anonymity and the ease of avoiding emotionally difficult conversations are key factors promoting ghosting behaviors. We also explore the psychological impacts of ghosting on both ghostees and ghosters, suggesting that, for many, the hurt of being rejected and the guilt of being responsible for this form of silent rejection have adverse consequences.

## 2. Psychological Factors

Persons with dark triad traits (i.e., psychopathy, Machiavellianism, and narcissism) tend to avoid the difficult conversations inevitably associated with ending a relationship. These individuals characteristically lack empathy with others and avoid pain for themselves whenever possible [1,3].

It has been found that the grandiosity and arrogance of many narcissists make them unwilling to admit errors or face conflict [4]. Machiavellian traits include duplicity; such

individuals often carry out multiple relationships at the same time, which may, from time to time, require dropping the one(s) that matter least. Psychopathy is characterized by dispositional callousness, which predisposes individuals to ghosting [1,3].

A study by Jonason et al. showed that men (more so than women) with dark triad traits find ghosting acceptable in short-term (as opposed to long-term) relationships [1]. Moreover, men with Machiavellian and psychopathic traits (but not those with narcissistic traits) report having ghosted people in the past and considering this form of ending relationships acceptable [1]. A study commented on this difference by speculating that narcissists are capable of empathy, whereas psychopaths and Machiavellians are not [3].

LeFebvre et al. suggested that ghosting is commonly used for terminating casual rather than serious relationships but does not comment on the association with dark triad traits [5]. Jonason et al. concluded that ghosting depends on one's concept of the seriousness of the relationship; the more a person values the relationship, the less likely they are to resort to ghosting when contemplating a breakup [1,3,5].

A psychological understanding of ghosting requires examining interpersonal dynamics through diverse lenses. Attachment theory offers one perspective, as attachment anxiety correlates positively with a need for constant validation, while an important risk for attachment avoidance is a history of unexplained intimacy withdrawal [21]. Both attachment anxiety and avoidance are theorized as driving ghosting behaviors, as a means, perhaps, of upholding a relatively fragile self-image [6,21]. Social interdependence theory provides another useful perspective [4,22]. Individuals focused on self-interest rather than on cooperation appear prone to suddenly disengage from relationships [22]. Personality traits are enhanced by digitally mediated communication channels that provide instant gratification but weak interpersonal ties [4]. The ghosting literature argues that collective passive avoidance dulls social accountability [5].

Comparisons have been made between ghosting and traditional face-to-face intimate breakup situations with respect to timing/duration, explanation/reasons, post-breakup contact, initiator status, relationship stage, social reactions, and related psychological states [5,23]. Ghosting is more likely to occur earlier in a courtship stage at a time when no clear commitment signals have been given by either party. Ghosting lacks direct closure conversations and, thus, elicits relatively more intense negative reactions.

### 3. Impact of Ghosting

Ghosting can affect both parties—ghostees and ghosters. Koessler et al. were of the opinion that young adults who initiated the termination process experienced less distress than those on the receiving end. Paradoxically, individuals who ended a relationship by speaking directly to their partner reported greater distress than did those who ghosted [18]. Another study conducted by Manning et al. revealed that individuals view the impact of ghosting differently depending on whether they are a ghoster or a ghostee [24]. The same individuals expressed the belief that having been ghosted was unjustifiable, but that ghosting was, nevertheless, a practical and simple way of ending a relationship [18].

Freedman et al. studied 80 individuals who had both ghosted and had been ghosted (38.8% men, 60% women, and one who did not specify sex; 10.0% African American, 8.8% Asian or Asian American, 5.0% Hispanic/Latinx, 1.3% Native American, 62.5% White, 8.8% Multiracial, 1.3% "other", and two persons who did not identify race/ethnicity) [2]. In other studies, there was no overall difference between ghosters and ghostees, in contrast to many papers that agreed that ghostees were the ones most negatively affected [18,25–27]. Ghosters experienced guilt but also relief, whereas ghostees were more likely to experience sadness and hurt [2,17]. Fundamental human needs such as belongingness, control, self-esteem, and meaningfulness of relational bonds were undermined in ghostees. The ghosters also reported feeling sad, angry, lonely, and frustrated, but less so if the relationship was brief. Ghostees also felt less hurt when the relationship was brief [2]. Uncertainty and self-doubt are some of the undesirable impacts resulting from ghosting. Individuals who are ghosted can start questioning their self-worth and overthinking the reasons behind

the ghosting to the point of rumination and sleep loss. As a result, they become cautious and hesitant about future relationships and lower their expectations of future potential connections [28].

This long-term impact was a concern of Jahrami et al. who considered the longer-term impact of ghosting on ghostees [4], who could, in the future, fear renewed intimacy or closeness in relationships. Sherry Turkle, a professor from MIT, stated in an interview for the Huffington Post that ghosting could have serious implications for ghostees' self-esteem and self-stigmatizing attitudes [29]. Jennice Vilhauer, on the Psychology Today website, is of the opinion that mental health professionals view ghosting as a form of emotional cruelty, leaving ghostees ignorant as to why a breakup occurred and powerless to respond [30].

Ghosting can occur in relationships other than those of friendship and romance, one such being that of a patient and psychotherapist [4]. Psychiatric ghosting (termination by a psychiatrist without discussion) is considered an unethical act that, though sometimes understandable, violates ethical, clinical, and termination guidelines [9,31]. As a consequence of such action, ghosted patients experience disbelief, annoyance, anxiety, profound sorrow, shame, and, at times, guilt. Being ghosted by someone in whom one has put one's trust induces intense reactions that may delay and impair involvement with future mental health providers. One unexpected finding is that, regardless of the negative emotions experienced by patients after they are ghosted, they continue to perceive their therapist as a good person [13].

Ghosting is associated with breadcrumbing, which is the act of sending out flirtatious texts merely for the sake of attention and without serious intent. Research by Navarro et al. studied 626 adults (303 males and 323 females) aged 18 to 40 years who had experienced ghosting and breadcrumbing [14]. Breadcrumbing, or flirtation without follow-up, was associated with diminished satisfaction in life and feelings of helplessness and loneliness, a correlation not found for ghosting. The explanation may lie in the direction of the effects. It is possible that lonely and dissatisfied people do not expose themselves to ghosting but may inadvertently become targets of breadcrumbing. It is well known that some individuals are much more sensitive to rejection than others [14,15].

Interestingly, the consequences of ghosting extend beyond psychology and can affect physiological processes. Cooper et al. and Fisher et al. reviewed fMRI studies depicting activated pain networks coinciding with interpersonal rejection [32,33]. Van der Molen et al. concluded that this activation occurs most often when a relationship ends abruptly [16], but any type of rejection can visibly impact the brain [34,35].

Although the majority of the published literature views ghosting from a negative perspective, it is worth noting that it might, in some instances, be linked with positive outcomes. When individuals invest in a relationship, they tend to undermine the toxic behavior exerted by the other party. In that sense, they step on their dignity to maintain ties and prolong the relationship even though it drains the energy from them. Consequently, after having been ghosted, ghostees are given the chance to focus on themselves and prioritize their emotional wellbeing, which enables the evolution of their understanding of relationships, bearing in mind all the do's and don'ts in addition to learning from any potential mistakes and growing as an individual. While going through the healing process, individuals tend to reflect back on their previous experiences and learn from them. With that, they learn to value their worth and to disallow those who are not worthy of free access to their life. This serves as guidance in the persuasion of future relationships by helping individuals become cautious when dealing with others and setting higher standards at many times to protect themselves and to prevent the recurrence of unsuccessful human contact. Moreover, they become action oriented, trusting actions more than words to preserve their time and effort invested in this connection at all costs. With this in the back of their mind, they seek to avoid any relationship with a partner that would bring about misery, self-destruction, insecurity, or hurt and rather aim for relationships that bring joy, happiness, calmness, self-love, comfort and appreciation. Instead of investing energy in

short-lived relationships, this energy could be diverted elsewhere to other aspects of life, such as personal growth, self-love, hobbies, or the refusal of other meaningful connections.

Ghosting itself is not easy to overcome because of the pain that results in varying intensities and severities depending on the intimacy of the relationship. This makes the ghostee aware that the other party is neither interested nor willing to put in efforts to maintain the relationship. Perhaps the termination of such relationships might be for the good. Over time, ghosting can teach people how to cope and live with rejection. It consolidates that we do not necessarily get everything we desire, as not everything we want is meant to be ours. In other words, its fate drives such individuals to go through heartbreak to learn how to address future encounters and, at the end of the day, helps them filter out those individuals who are not compatible with their own values, communication styles, or relationship expectations. This could ultimately lead to finding more compatible and fulfilling connections. It is crucial to note that these benefits might not be applicable in all situations. Just like how one shoe does not fit all, every person and relationship is unique. Furthermore, the consequences of ghosting can vary depending on personal factors. This is where supporting friends and family members come into play to aid in easing the aftermath of ghosting.

Our review reveals that ghosting proclivity is concentrated among individuals exhibiting dark triad traits, such as narcissism and Machiavellianism, suggesting that these personality facets cultivate callous tendencies. Rather than online disinhibition through anonymity stripping social guardrails, as prominently argued, the data indicate that detachment and cowardly behavior in interpersonal situations have existed long before the digital revolution. Technology simply grants plausible deniability for innate propensities. In turn, normalized ghosting risks a feedback loop that reinforces selective empathy and erodes accountability and communal bonds.

## 4. Cultural and Societal Factors

Technology has made it possible to interact with individuals in all corners of the world with a single click. It has become feasible to link oneself with a partner through online dating apps and websites and stay in constant touch over the miles by texting and video calls [2,17]. Currently, mobile applications are the most common method for seeking romance; traditional introductions and meeting strategies have dramatically decreased worldwide [4].

The initiation of online dating can be traced back 20–25 years, and dating apps are used by young adults but are becoming increasingly popular in all age groups. Dating apps provide many options, allowing individuals to gather information about each other before meeting in person and permitting prior negotiation of a relationship. The downside is the emergence of behaviors such as "breadcrumbing", "slow fading", "benching", "haunting", or "ghosting"—new names for flirting, deceiving, initiating, maintaining, or terminating relationships, often on a whim [4,14,15].

The wide accessibility of dating apps has decreased the intimacy of in-person interactions. A new phenomenon is "online vigilance", which refers to constant preoccupation with the online world even when offline [36]. The constant availability of alternative dating partners promotes ghosting without guilt [28].

The advent of artificial intelligence is another technological advancement that can promote ghosting. Fake profiles and bots are prevalent on dating apps, and when a request for an in-person meeting is made, the scammer may then resort to ghosting. Research is needed on chatbot-generated communication on dating apps, how such communication facilitates chatting and how this form of connection correlates with ghosting.

Significant avenues for further exploration of the ghosting construct remain. Future work could benefit from extending the age ranges, cultural backgrounds, and technological contexts assessed here. The population addressed in the ghosting literature is currently skewed heavily towards young adults from high income Western countries. Older cohorts may experience online abandonment differently, and the motivations behind it may differ in

older individuals. The impacts and motivations of ghosting may also vary cross-culturally, based on norms of courtesy and confrontation avoidance, as well as the importance of saving face and the value of individualism. Future work should embrace diverse samples and communication platforms, such as social media, gaming, and virtual worlds. Expanding the variables would clarify the generalizability of the effects. Overall, this research provides an empirical foundation from which to systematically address ghosting antecedents and externalities through rigorous hypothesis testing.

## 5. Coping Strategies

Most people learn to cope with rejections, but it is important to remind patients in therapy that ghosting is not a reflection on them but rather on the personal traits of the ghoster [4,14,15]. This approach is believed to constitute a passive-aggressive method of ending relationships. Individuals who repeatedly ghost others should be advised to examine their motivations because of personality traits that bode ill for the establishment of long-term intimacy [4,14,15].

Timmermans et al. reported several ways in which ghostees cope with ghosting. The most commonly mentioned coping mechanism was rationalizing that it was an inevitable product of online dating, but some respondents opted to delete the dating app. Others found that sharing their rejection story with friends was helpful. Some tried to reach out to the ghoster through social media to comprehend the cause behind the absence of contact, and it was then that they realized that they had been ghosted [14]. Some researchers have attempted to find justification for this relationship failure [28]. Some ghostees stated that this experience would change their future behavior and that they might now turn into ghosters [14].

With respect to therapist ghosting, some ghostees tried to reach out to their therapists for an explanation, but most (83.9%) did not, concluding either that they were somehow responsible (21.3%) or that something occurred in the therapist's personal life. Approximately 16% of the patients in this study managed to recontact their therapist [13].

Future studies assessing the emotional/psychological consequences of ghosting should utilize validated clinical scales to evaluate outcomes such as anxiety, depression, self-esteem, and feelings of rejection before and after ghosting. Standardized measures allow reliable quantitative analysis of the directionality and magnitude of ghosting effects. Qualitative data obtained through interviews and self-reports could also provide nuanced information about lived experiences. Moreover, research examining coping mechanisms must carefully define and quantify interventions, tracking efficacy through the use of validated assessment instruments. Self-reports can complement but should not serve as the sole measure of efficacy, as subjective perceptions of improvement are not reliable. The use of control groups is needed to help account for natural recovery over time, irrespective of coping interventions.

## 6. Conclusions and Prospects

Ghosting, the abrupt and unexplained cessation of communication in relationships, has become increasingly prevalent in today's digital era. It occurs across various types of relationships, including romantic, platonic, and professional connections. The consequences of being ghosted can be profound, causing emotional distress, confusion, and damage to one's self-esteem and trust in others. It is important to note that ghosting is not meant to do harm but is most likely intended to avoid difficult conversations for the ghoster.

Through research, we can more thoroughly explore the motives and intentions that drive ghosters to abruptly cease communication. Factors such as the fear of confrontation, desire for control, or emotional immaturity may be involved.

Perhaps more important is the investigation of the emotional and psychological consequences that ghostees face. This means exploring feelings of rejection, abandonment, and self-doubt. Such exploration prevents future difficulties in establishing future relationships. Additionally, studying potential variations in the impact of ghosting based

on factors such as attachment style, gender, and cultural background will contribute to improved psychotherapeutic techniques. Gaining insight into how individuals can use coping mechanisms (self-care, resilience, healthy communication) to recover and heal from the experience of being ghosted can significantly contribute to the development of targeted interventions and support systems. Another possibility is to explore preventive strategies that can be implemented inside social media to prevent or at least reduce the prevalence of ghosting. This is a new and promising field of mental health prevention.

Our review reveals several insights that could guide the design of instant messaging, dating apps, and digital platforms to help mitigate the adverse effects of ghosting behaviors. For instance, we find associations between ghosting tendencies and dark personality traits such as narcissism. This suggests that dating apps could potentially screen for these traits or incorporate notifications/tips that discourage ghosting by appealing to users' empathy. Additionally, finding that features such as asynchronous messaging enable avoidance of difficult conversations implies that apps could require synchronous communication after a certain messaging threshold, reducing ghosting prevalence.

More broadly, results pointing to links between ghosting and emotional immaturity or need for control means that apps could try to promote personal growth and vulnerability. For example, apps might gamify self-reflection prompts for matches to complete together, facilitating intimate conversation. Or they could encourage perspective-taking measures by having users annotate profiles with what they appreciate in a potential partner. Features that cultivate emotional intelligence may curb the propensity to ghosting.

Moreover, our findings regarding the deep interpersonal wounds inflicted by ghosting indicate a need for apps to build ghosting recovery tools. For instance, if a user stops responding, the app could send gentle nudges, followed by a closure message if radio silence persists. Users who were ghosted could be directed to optional counseling resources. A mentorship community may also help users process feelings of rejection post ghosting. By acknowledging ghosting harms, apps can change cultural norms.

**Author Contributions:** Conceptualization, L.R.D. and H.J.; methodology, L.R.D. and H.J.; software, not applicable; validation, H.J.; formal analysis, L.R.D. and H.J.; investigation, L.R.D. and H.J.; resources, H.J.; data curation, not applicable; writing—original draft preparation, L.R.D., H.J., and M.V.S.; writing—review and editing, L.R.D., G.P., H.J., M.V.S., and M.R.B.; visualization, L.R.D., H.J., and M.R.B.; supervision, H.J.; project administration, H.J.; funding acquisition, H.J. All authors have read and agreed to the published version of the manuscript.

**Funding:** This research received no external funding.

**Institutional Review Board Statement:** This study did not require ethical approval.

**Informed Consent Statement:** Not applicable.

**Data availability statement:** No new data were created.

**Conflicts of Interest:** GP is the owner of the online support company "Coping with Ghosting LLC". The subject matter of this manuscript did not have any direct bearing on that work, nor has that activity exerted any influence on this project. All authors declare that the research was conducted in the absence of any commercial or financial relationships that could be construed as a potential conflict of interest.

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
