# Peer review of "Ghosting: Abandonment in the Digital Era"

_encyclopedia, doi:10.3390/encyclopedia4010004_

Round 1
Reviewer 1 Report
Comments and Suggestions for Authors
have carefully reviewed your manuscript entitled "Ghosting: Abandonment in the Digital Era," which addresses the phenomenon of ghosting in the context of modern technology and dating applications. This letter contains my recommendations for significant revisions before the manuscript can be considered for publication.
Abstract and Introduction: The abstract succinctly summarizes the rise of ghosting alongside the advent of technology. However, the introduction would benefit from a broader context of social relationships and technological impact over time. It should also outline the central research question more explicitly and provide a clear thesis statement.
Literature Review: The review of literature connecting dark triad personality traits to ghosting is intriguing, yet it seems narrow. Consider expanding the review to include broader psychological theories that may contribute to this behavior. Additionally, comparative analyses with traditional forms of relationship termination could provide a richer background.
Methodology: The methodology section is not present in the abstract but is crucial for assessing the validity of the research. It should detail the methods used to gather data on ghosting, the operational definitions of ghosters and ghostees, and the criteria for assessing the psychological impact of ghosting.
Data Analysis and Interpretation: The abstract mentions that most individuals find ways of coping with the psychological effects of ghosting. However, the manuscript should discuss the methodologies used to measure mental health impacts and coping mechanisms. Ensure that the data analysis is thorough and that the interpretation aligns with the results.
Discussion of Results: The discussion should integrate findings with existing literature, exploring how the study confirms, challenges, or extends current understanding. It should also acknowledge limitations, such as potential biases in self-reporting and the representativeness of the sample.
Theoretical Implications: Your findings about the dark triad and ghosting could have significant theoretical implications. Delve deeper into how these results contribute to personality psychology and our understanding of interpersonal relationships in the digital age.
Practical Implications: While the manuscript alludes to practical implications, it would be strengthened by a clear articulation of how these findings can be applied. For instance, what do these results suggest for the design of dating apps to mitigate the negative consequences of ghosting?
Originality and Significance: The topic is undoubtedly timely and relevant. To highlight the originality, compare your work with existing studies and clarify what new insights your study provides. What does your research offer that moves beyond the current discourse. Your conclusion should not only summarize the findings but also suggest future research directions based on the study's limitations and findings.
In summary, your manuscript offers an insightful look into a modern phenomenon that is impacting social and romantic relationships. The revisions I have recommended aim to deepen the analysis and broaden the manuscript's scope, which will make it a valuable contribution to the literature on interpersonal relationships in the digital era.
Comments on the Quality of English Language
No problem
Author Response
Responses to Reviewer 1
I have carefully reviewed your manuscript entitled "Ghosting: Abandonment in the Digital Era," which addresses the phenomenon of ghosting in the context of modern technology and dating applications. This letter contains my recommendations for significant revisions before the manuscript can be considered for publication.
Abstract and Introduction: The abstract succinctly summarizes the rise of ghosting alongside the advent of technology. However, the introduction would benefit from a broader context of social relationships and technological impact over time. It should also outline the central research question more explicitly and provide a clear thesis statement.
Authors’ response: We thank you for your careful review of our manuscript "Ghosting: Abandonment in the Digital Era" and for your suggestions on improving the abstract and introduction. We appreciate you taking the time to provide feedback to strengthen our paper.
In response to your comments, we have revised the introduction to provide more context about the historical evolution of social relationships alongside technological developments. The goal was to illustrate how modes of communication and relating to others have shifted over time with each new advancement, leading up to the current emergence of ghosting through dating apps and digital communication platforms. We added the following paragraphs:
“The emergence of ghosting must be understood within the shifting of historical dynamics between direct person-to-person relating and the new mode of relating via communication media [5]. Face-to-face meetings are what used to build and strengthen social ties, making each party accountable for behaviour during contact [10]. The advent of telegraph and telephone subsequently permitted communication during physical separations, thus maintaining relationships over a distance, but reducing closeness [11]. Radio and television centralized one-way intimate broadcasts into homes, creating fantasy relationships that, at times, felt real [4]. Internet forums, emailing, and text messaging allowed fast, low-cost, asynchronous connections between people over geographical distance, spurring a form of disembodied intimacy among relative strangers [5]. Social platforms have expanded this form of pseudo-intimacy [9, 12]. Each advancement, perhaps paradoxically, made it easier to avoid the difficult conversations that inevitably arise over time between persons who are building a potentially intimate relationship [10]. The present day practice of ghosting thus reflects what happens when attempting to relate through impersonal means rather than face-to-face. It takes advantage of the impersonal to avoid painful conversations and social accountability.”
“Dating apps are the latest communication technology to affect relationships. Features such as swiping and messaging foster quick superficial judgments of the persons with whom one “talks.” This makes connections relatively remote and fosters the emergence of ghosting, seemingly painless break ups [12].”
Additionally, we have clarified our central research question, as follows: “Our central research question in this entry examines ghosting motivations and their effects on the persons involved. We propose that technologically-induced anonymity and the ease of avoiding of emotionally difficult conversations are key factors promoting ghosting behaviours. we also explore the psychological impacts of ghosting on both ghostees and ghosters, suggesting that, for many, the hurt of being rejected and the guilt of being responsible for this form of silent rejection have adverse consequences.”
Literature Review: The review of literature connecting dark triad personality traits to ghosting is intriguing, yet it seems narrow. Consider expanding the review to include broader psychological theories that may contribute to this behavior. Additionally, comparative analyses with traditional forms of relationship termination could provide a richer background.
Authors’ response: We added the following brief/broad summary of the main psychological theories that contribute to ghosting. Additionally, we provided brief comparative analyses with traditional forms of relationship termination that now provide a richer background to the ghosting concept:
“A psychological understanding of ghosting requires examining interpersonal dynamics through diverse lenses. Attachment theory offers one perspective, as attachment anxiety correlates positively with a need for constant validation, while an important risk for attachment avoidance is a history of unexplained intimacy withdrawal [21]. Both attachment anxiety and avoidance are theorized as driving ghosting behaviours, as a means, perhaps, of upholding a relatively fragile self-image [6, 21]. Social interdependence theory provides another useful perspective [4, 22]. Individuals focused on self-interest rather than on cooperation appear prone to suddenly disengage from relationships [22]. Personality traits are enhanced by digitally mediated communication channels that provide instant gratification but weak interpersonal ties [4]. The ghosting literature argues that collective passive avoidance dulls social accountability [5].
Comparisons have been made between ghosting and traditional face-to-face intimate breakup situations with respect to timing/duration, explanation/reasons, post-breakup contact, initiator status, relationship stage, social reactions, and related psycho-logical states [5, 23]. Ghosting is more likely to occur earlier in a courtship stage at a time when no clear commitment signals have been given by either party. Ghosting lacks direct closure conversations, and, thus, elicits relatively more intense negative reactions.
Methodology: The methodology section is not present in the abstract but is crucial for assessing the validity of the research. It should detail the methods used to gather data on ghosting, the operational definitions of ghosters and ghostees, and the criteria for assessing the psychological impact of ghosting.
Authors’ response: We added the following to the abstract: “This entry synthesizes the multidisciplinary literature on ghosting published through late 2023 across psychological and social science journals. Search terms include “ghosting” and “online dating.” Quantitative and qualitative studies are included.”
Data Analysis and Interpretation: The abstract mentions that most individuals find ways of coping with the psychological effects of ghosting. However, the manuscript should discuss the methodologies used to measure mental health impacts and coping mechanisms. Ensure that the data analysis is thorough and that the interpretation aligns with the results.
Authors’ response: We added the following paragraph: “Future studies assessing the emotional/psychological consequences of ghosting should utilize validated clinical scales to evaluate outcomes such as anxiety, depression, self-esteem, and feelings of rejection before and after ghosting. Standardized measures allow reliable quantitative analysis of the directionality and magnitude of ghosting effects. Qualitative data obtained through interviews and self-reports could also provide nuanced information about lived experiences. Moreover, research examining coping mechanisms must carefully define and quantify interventions, tracking efficacy through the use of validated assessment instruments. Self-reports can complement but should not serve as the sole measure of efficacy, as subjective perceptions of improvement are not reliable. The use of control groups is needed to help account for natural recovery over time, irrespective of coping interventions.”
Discussion of Results: The discussion should integrate findings with existing literature, exploring how the study confirms, challenges, or extends current understanding. It should also acknowledge limitations, such as potential biases in self-reporting and the representativeness of the sample.
Authors’ response: We added the following paragraph: “Significant avenues for further exploration of the ghosting construct remain. Future work could benefit from extending the age ranges, cultural backgrounds, and technological contexts assessed here. The population addressed in the ghosting literature is currently skewed heavily towards young adults from high income Western countries. Older cohorts may experience online abandonment differently and the motivations behind it may differ in older individuals. The impacts and motivations of ghosting may also vary cross-culturally, based on norms of courtesy and confrontation avoidance, as well as the importance of saving face and the value of individualism. Future work should embrace diverse samples and communication platforms, such as social media, gaming, and virtual worlds. Expanding the variables would clarify the generalizability of the effects. Overall, this research provides an empirical foundation from which to systematically address ghosting antecedents and externalities through rigorous hypothesis testing.”
Theoretical Implications: Your findings about the dark triad and ghosting could have significant theoretical implications. Delve deeper into how these results contribute to personality psychology and our understanding of interpersonal relationships in the digital age.
Authors’ response: We added the following paragraph to further expand the discussion about the link between the ‘dark triad’ and ghosting, as follows: “Our review reveals that ghosting proclivity is concentrated among individuals exhibiting dark triad traits, such as narcissism and Machiavellianism, suggesting that these personality facets cultivate callous tendencies. Rather than online disinhibition through anonymity stripping social guardrails, as prominently argued, the data indicate that detachment and cowardly behaviour in interpersonal situations have existed long before the digital revolution. Technology simply grants plausible deniability for innate propensities. In turn, normalized ghosting risks a feedback loop that reinforces selective empathy and erodes accountability and communal bonds.”
Practical Implications: While the manuscript alludes to practical implications, it would be strengthened by a clear articulation of how these findings can be applied. For instance, what do these results suggest for the design of dating apps to mitigate the negative consequences of ghosting?
Authors’ response: We provided the following arguments to strengthen the practical implications: “Our review reveals several insights that could guide the design of instant messaging, dating apps and digital platforms to help mitigate the adverse effects of ghosting behaviors. For instance, we find associations between ghosting tendencies and dark personality traits such as narcissism. This suggests that dating apps could potentially screen for these traits or incorporate notifications/tips that discourage ghosting by appealing to users' empathy. Additionally, finding that features such as asynchronous messaging enable avoidance of difficult conversations implies that apps could require synchronous communication after a certain messaging threshold, reducing ghosting prevalence.
More broadly, results pointing to links between ghosting and emotional immaturity or need for control means that apps could try to promote personal growth and vulnerability. For example, apps might gamify self-reflection prompts for matches to complete together, facilitating intimate conversation. Or they could encourage perspective-taking by having users annotate profiles with what they appreciate in a potential partner. Features that cul-ivate emotional intelligence may curb the propensity to ghosting.
Originality and Significance: The topic is undoubtedly timely and relevant. To highlight the originality, compare your work with existing studies and clarify what new insights your study provides. What does your research offer that moves beyond the current discourse. Your conclusion should not only summarize the findings but also suggest future research directions based on the study's limitations and findings.
Authors’ response: We added the following paragraph: “Moreover, our findings regarding the deep interpersonal wounds inflicted by ghosting indicate a need for apps to build ghosting recovery tools. For instance, if a user stops responding, the app could send gentle nudges, followed by a closure message if radio silence persists. Or users who were ghosted could be directed to optional counseling resources. A mentorship community may also help users process feelings of rejection post-ghosting. By acknowledging ghosting harms, apps can change cultural norms”.
In summary, your manuscript offers an insightful look into a modern phenomenon that is impacting social and romantic relationships. The revisions I have recommended aim to deepen the analysis and broaden the manuscript's scope, which will make it a valuable contribution to the literature on interpersonal relationships in the digital era.
Authors’ response: We sincerely appreciate you taking the time to thoroughly review our manuscript and provide constructive feedback on enhancing the depth and breadth of our analysis on ghosting behaviors. We feel that the resulting changes have greatly strengthened the manuscript, and we hope they will result in a favorable decision to publish our work in encyclopedia MDPI.

Reviewer 2 Report
Comments and Suggestions for Authors
The text presented for the review consists of an abstract, five sections, and references. The text addresses the phenomenon of ghosting, specifically how this phenomenon differs, depending on whether it pertains to friendships or romantic relationships. The text includes numerous research findings and references to works of other researchers in this field. From the many references to these sources, information emerges about models of reactions to ghosting, both on a social and cultural level, as well as on a physiological level. The text can go to press.
Author Response
Responses to Reviewer 2
The text presented for the review consists of an abstract, five sections, and references. The text addresses the phenomenon of ghosting, specifically how this phenomenon differs, depending on whether it pertains to friendships or romantic relationships. The text includes numerous research findings and references to works of other researchers in this field. From the many references to these sources, information emerges about models of reactions to ghosting, both on a social and cultural level, as well as on a physiological level. The text can go to press.
Authors’ response: Thank you very much for your positive feedback on our manuscript titled "Ghosting: Abandonment in the Digital Era". We are thrilled to hear that you found the subject matter interesting and relevant to the current times.
No further action was needed.
Reviewer 3 Report
Comments and Suggestions for Authors
The authors did a great job focusing on an interesting, contemporary and current subject for our time: ghosting - abandonment in the Digital Era.
As an entry manuscript, the review of the studies is clear and gives a good background, while delving into the definition of the term, as well as the psychological factors, the impact of ghosting, cultural and societal factors, and coping strategies.
I really enjoyed reading it!
Author Response
Responses to Reviewer 3
The authors did a great job focusing on an interesting, contemporary and current subject for our time: ghosting - abandonment in the Digital Era.
As an entry manuscript, the review of the studies is clear and gives a good background, while delving into the definition of the term, as well as the psychological factors, the impact of ghosting, cultural and societal factors, and coping strategies.
I really enjoyed reading it!
Authors’ response: Thank you very much for your positive feedback on our manuscript titled "Ghosting: Abandonment in the Digital Era". We are thrilled to hear that you found the subject matter interesting and relevant to the current times.
No further action was needed.
